# Beyond the Greater Angkor Region: Automatic large-scale mapping of Angkorian-period reservoirs in satellite imagery using deep learning

**Jürgen Landauer**[1], **Sarah Klassen**[2,3]*, **Adam P. Wijker**[4], **Josine van der Kroon**[5], **Alexis Jaszkowski**[1], **Wouter Baernd Verschoof-van der Vaart**[5]

**1** Landauer Research, Ludwigsburg, Germany, **2** Archaeology Centre, University of Toronto, Toronto, Ontario, Canada, **3** Institute of Behavioral Science, University of Colorado Boulder, Boulder, Colorado, United States of America, **4** École française d'Extrême-Orient (EFEO), Université Paris I Panthéon-Sorbonne, Paris, France, **5** Faculty of Archaeology, Leiden University, Leiden, Netherlands

* sarahe.klassen@utoronto.ca

## Abstract

Archaeologists often use high-resolution satellite imagery to identify potential archaeological sites or features, including ancient settlements, burial mounds, roads, and even subtle differences in vegetation or topography. Over the last several decades, satellite imagery and other remote sensing techniques (including aerial photography and LiDAR data) have been used to thoroughly map the extensive settlement complex of the Greater Angkor Region (1 500 km², 9th – 14th centuries CE) in present-day Cambodia. While we now have a comprehensive map of this area, the landscapes beyond the Greater Angkor Region that formed the Angkorian cultural sphere have not been mapped, even though the density of features on the landscape seems to continue beyond the area considered Greater Angkor. While a comprehensive settlement study of the entire Angkorian realm would be incredibly helpful in understanding patterns of ancient urbanism and early statehood in Southeast Asia, mapping this area using manual identification of archaeological features in satellite imagery would be highly time-consuming. In this paper, we employ a state-of-the-art deep learning model for semantic segmentation using Deeplab V3+ to identify one typical and characteristic feature: Angkor-period reservoirs. Our results indicate that this AI model is accurate enough to provide a valuable "second opinion" to landscape archaeologists to enhance and quicken their mapping process, making them substantially more productive. The deep learning model for semantic segmentation employed here, which can be trained on other types of archaeological and non-archaeological features worldwide, will be a valuable tool for areas of research that involve intensive manual investigation and interpretation of satellite imagery and will aid researchers as they continue to map the Angkorian world.

**Data availability statement:** The source code of this project is publicly available on Github (https://github.com/juergenlandauer/cambodia-khmer-semantic-segmentation) along with usage instructions. The underlying satellite imagery is available via the Microsoft Bing Maps API (see https://learn.microsoft.com/en-us/bingmaps/rest-services/). The Microsoft Bing Maps Platform APIs' terms of use are available at https://www.microsoftcom/en-us/maps/product/print-rights. The data are not publicly available due to ethics of archaeological site protection. The site locations used for training and testing may be available for research purposes, with appropriate request and have been archived at tdar (tDAR id: 501902, or see doi:10.48512/XCV8501902). The satellite imagery used in this article is provided by the Planet Labs Open Data programme (https://www.planet.com/data/stac/browser) and Sentinel-2 cloudless (https://s2maps.eu) by EOX IT Services GmbH, both available under a Creative Commons Attribution-NonCommercial-ShareAlike 4.0 International License.

**Funding:** European Union's Horizon 2020 Marie Sklodowska-Curie grant agreement No. 896092 (sk), Fellowship support for the University of Toronto (sk).

**Competing interests:** The authors have declared that no competing interests exist.

# 1. Introduction

Archaeologists frequently use satellite imagery and other remote sensing techniques to identify and map archaeological features [1–3]. Satellite imagery can provide valuable information about the Earth's surface, as the aerial perspective allows archaeologists to identify features not easily visible from the ground, and archives of past imagery provide time depth. Similarly, satellite imagery can be used to detect variations, such as crop marks or soil marks, that may indicate the presence of buried structures or other archaeological remains [4,5]. From satellite and other remote sensing imagery, archaeologists have made detailed maps and documented thousands of archaeological sites in the Greater Angkor Region [6,7], which has allowed researchers to plan excavations and conduct further studies [8].

Our case study, the Angkorian world, was a powerful and influential civilization that flourished in Southeast Asia from approximately the 9th to 15th centuries C.E. As conventionally defined from epigraphic sources, the Angkor Period began in 802 C.E. when King Jayavarman II united disparate kingdoms into one polity [9,10]. Over the next several centuries, it expanded its territory to encompass present-day Cambodia and parts of Thailand, Laos, and Vietnam, making it one of the most extensive empires in Southeast Asia at the time [10]. It is best known for its principal city, Angkor, and the magnificent Angkor Wat temple complex, one of the world's most iconic and well-preserved archaeological sites. Using a combination of remote sensing data, including satellite and radar imagery and LiDAR data, archaeologists have mapped over 25 000 archaeological features in the Greater Angkor Region (see below) [7]. This work was the culmination of decades of regional research [11,12]. In 2012 and 2015, a series of LiDAR surveys across Cambodia revealed extensive landscape manipulation that extended beyond the boundaries of the Greater Angkor Region and may have stretched across much of the part of mainland southeast Asia that was once part of the empire [13]. While many of the densely occupied urban centers, sometimes referred to as civic-ceremonial zones [7], beyond Greater Angkor have now been mapped as a result [14,15], it would be extremely costly and time-consuming to map the tens of thousands of features that connect these other centers with Angkor using manual methods alone. As such, automated methods are vital for addressing questions regarding the extent of landscape modification in the Angkorian world.

This paper briefly reviews the mapping work done in the Greater Angkor Region and across Cambodia. We also explain the necessity and challenges of mapping archaeological features such as temples and reservoirs across substantially larger geographic regions than have hitherto been covered. One of the primary challenges is the sheer amount of time required to manually identify thousands of archaeological features in satellite imagery across broad landscapes or even at a country or region-wide scale. Based on this high time demand, an AI algorithm may assist archaeologists with this task and reduce the man-hours required. The following sections provide details concerning the data, methods, and AI techniques we employed for semantic segmentation. We then discuss the results we obtained in our test areas, which suggest that the method can significantly reduce the amount of labor required to survey a given area.

# 2. Background

## 2.1. Background to the Greater Angkor Region and research problem

The Greater Angkor Region was home to several successive capitals of the Khmer Empire from the 9th to 15th centuries C.E. [16]. The region is situated in the low-lying floodplain of the Tonle Sap lake and river, which are part of the Mekong River system [17]. The annual monsoon rains cause the Tonle Sap river to reverse its flow, causing water levels in the lake to rise and providing fertile sediment-rich soil for agriculture [18]. The people of Angkor capitalized

on this natural resource and transformed the floodplain into an extensive network of rice fields, reservoirs, and canals interwoven among domestic and ritual architecture [19–21].

Angkor is a sprawling complex of over 1 000 temples and tens of thousands of occupation mounds connected through a network of hydraulic infrastructure, including channels and reservoirs [19,20,22]. These temples are associated with agricultural space and bunded rice fields, forming distinct agricultural communities. The reservoirs, of particular note for this paper, were built by each community and likely served multiple purposes, including as a source of protein (fish) and water for livestock, and as part of the ritual landscape. A typical agricultural temple community tends to have about three reservoirs clustered around the temple, with one directly connected to it by a causeway [20].

For their dissertations, Christophe Pottier and Damian Evans mapped approximately 1 000 temples and thousands of reservoirs in the southern and northern areas, respectively, of what has come to be defined as the Greater Angkor Region using aerial and satellite imagery [11,23]. It is important to note that the boundaries of the Greater Angkor Region are derived from the watersheds of its major rivers. While not completely arbitrary, the study area is thus determined by environmental factors rather than the known extent of archaeological remains [11]. While these methods worked well for mapping the "hinterlands" of Angkor, the site core could not be mapped due to the dense vegetation that obscured archaeological features from view (Fig 1).

In 2012, the Khmer Archaeological LiDAR Consortium (KALC) organized an airborne Light Detection and Ranging (LiDAR) acquisition over the densely vegetated areas of Angkor (red boxes in Fig 1). The results of the survey revealed a complex urban fabric of streets clearly outlining city blocks comprised of residential mounds and ponds [6]. Over 25 000 features have now been comprehensively mapped and ground verified by a group of researchers associated with the Greater Angkor Project, led by Evans and including two authors of this paper, Wijker and Klassen (Fig 2).

Based on the success of the 2012 survey, the Cambodian Archaeological LiDAR Initiative (CALI) commissioned a further set of surveys in 2015 [13]. They revealed several previously concealed and undocumented dense urban landscapes in Cambodia associated with large temple complexes (Fig 3). These LiDAR surveys were primarily centered on areas of dense

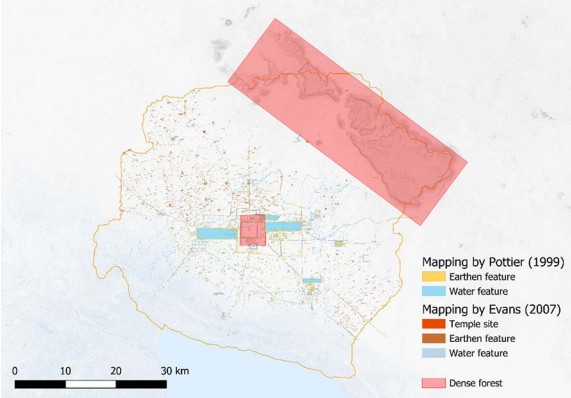

**Fig 1. Mapping of the North and South portions of the Greater Angkor Region by Pottier and Evans.** The red boxes highlight the areas that are difficult to map from satellite imagery due to dense vegetation. Image sources: Topographic data from the NASA Shuttle Radar Topography Mission (SRTM). Hydrographic data made with Natural Earth, overlay: authors, Pottier (1999) and Evans (2007).

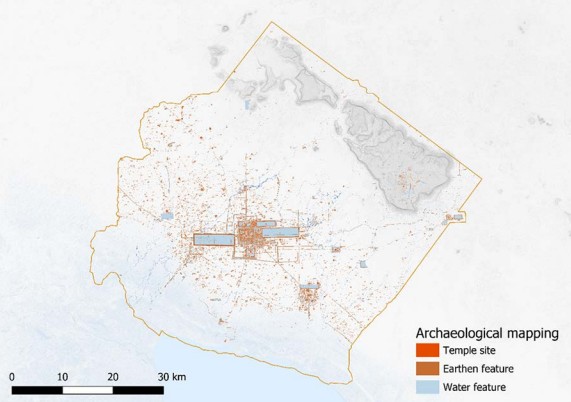

**Fig 2. Mapping of the LiDAR data.** Image sources: Topographic data from the NASA Shuttle Radar Topography Mission (SRTM). Hydrographic data made with Natural Earth.

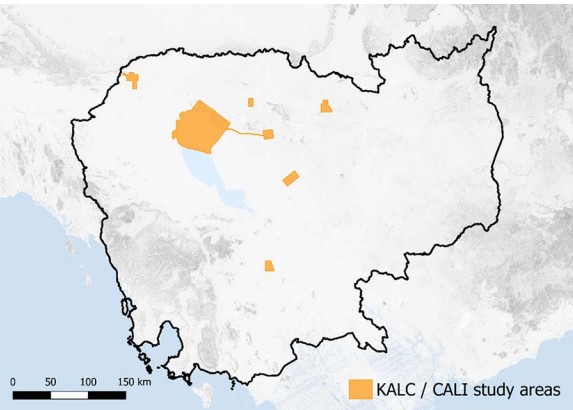

**Fig 3. Site locations surveyed as part of the Cambodian Archaeological LiDAR Initiative.** Image sources: Topographic data from the NASA Shuttle Radar Topography Mission (SRTM). Hydrographic data made with Natural Earth.

vegetation, and mapping of these areas has shown that, in many cases, archaeological features continue up to and beyond the bounds of the study area.

These extensive mapping projects have revealed two key insights relevant to this study:

1) The Greater Angkor Region has two distinct archaeological signatures of occupation. The first, identified and mapped by Pottier and Evans, is a low-density agricultural pattern comprised of small community temples, reservoirs, and rice fields [7,16]. The second, identified by Gaucher [24] and mapped with the aid of the LiDAR data, is a densely occupied "urban core," sometimes referred to as the "civic-ceremonial center," characterized by occupation mounds, monumental temples, and ponds with a distinct lack of rice fields. As many as 150 000 people may have lived in the civic-ceremonial center, and a further 500 000 - 700 000 may have lived in the low-density agricultural zone of the Greater Angkor Region at its height in the 13th century [7]. This pattern of more and distinctly less densely occupied agricultural areas seems broadly consistent at the other urban centers surveyed in Cambodia in 2015 [13].

2) The mapping reveals that the occupation density of the low-density agricultural zone did not drop off at the boundaries of the Greater Angkor Region and presumably extends well beyond. It may even extend up to and connect with the other urban centers across Cambodia identified and mapped by the CALI project.

These insights have critical implications for our understanding of the nature of Khmer settlement systems, the complexity of the agricultural system, and the nature of the rise of urbanism in Southeast Asia. Beyond documenting the urban layout of the densely occupied areas within each of the LiDAR zones, we can use this data to understand better the extent and boundaries of urban and agricultural areas and to estimate the historical population of those zones. Knowing how much water was stored in the associated reservoirs is also helpful in modeling the Angkorian agricultural system. However, before we can address these questions on a broader scale, we must first document and map archaeological features beyond the bounds of the Greater Angkor Region and the outlying urban centers.

## 2.2. Deep learning models: Semantic segmentation

The cultural heritage field has been slow to adopt Machine Learning (ML) methods [25]; however, archaeologists have successfully experimented with image classification, object detection, and segmentation to identify archaeological features in remotely sensed data [26–28]. These approaches complement other ML methods used by archaeologists. Examples include graph-based semi-supervised ML algorithms for determining archaeological site dates [22,29] and classification analyses for improved labeling and understanding of datasets and field reports [30,31]. Similarly, Region-based Convolutional Neural Networks (R-CNNs), a technique from the Deep Learning domain have been applied to LiDAR data to detect archaeological features like prehistoric barrows and Celtic fields in the Netherlands [32,33] and charcoal kilns, grave mounds, and pitfall traps in Norway [34]. For single-stage detectors, Yolov3 has been used to detect burial mounds in northwestern Iberia [35], while RetinaNet has been employed to pinpoint charcoal production sites in a wooded area of Jönköping County, Sweden [36]. Two recent studies applying ML in Mesopotamia and Uzbekistan as well as the Andes to develop a collaborative human-AI workflow that is similar to ours [37,38].

Outside of the archaeological domain, the detection of swimming pools and similar water bodies is a well-established field of research, ranging from academic studies [39] to the practical use of AI detection of pools by French tax authorities to combat tax evasion [40]. Additionally, many research teams, such as the participants in the SpaceNet 8 challenge in 2022 [41], have used automated detection to identify flooding in satellite imagery. These applications are very similar to those we used to identify ancient reservoirs. The key difference between these modern-day examples and our research question is the historical dimension, or more precisely, the need to consider the depositional histories of archaeological sites. As such, our problem is substantially more complex since we need to train our model to detect Angkor-era reservoirs even if they no longer contain any trace of water and if the retaining walls no longer exist due to erosion and destruction.

## 3. Materials and methods

### 3.1. Dataset creation: Satellite data from Microsoft Bing and Khmer reservoir shapes

We began by compiling pre-existing survey records. Data from the CISARK database, a collaboration between the École française d'Extrême-Orient and the Cambodian Ministry of Culture and Fine Arts (https://cisark.mcfa.gov.kh/), includes over 4 000 temple complexes beyond the

Greater Angkor Region. As mentioned above, temples in the Greater Angkor Region each have an average of three associated reservoirs. When we began this study, we expected to find at least 12 000 reservoirs across Cambodia. Upon manual investigation of satellite imagery from select areas of Cambodia, we soon found many more unrecorded temple complexes and thousands of reservoirs in the relatively small portions of the country we surveyed. Identifying and mapping all these features by hand would be an overwhelming task. We turned to automated techniques to help identify and, hopefully, map ancient temples and reservoirs on the landscape.

To train our ML model, we used satellite data of Cambodia from Microsoft Bing, which provides high-resolution imagery of up to 30 cm per pixel at low or no cost [42]. The only drawback is that the actual resolution for any given area is not precisely indicated, since Bing attempts to provide a worldwide, unified data source from various satellite sources without revealing the underlying resolution [43]. However, visual inspection of the data from Cambodia suggests that it falls in the 2 m per pixel range – a sufficiently high resolution for deep learning applications in our domain.

More than 11 000 reservoirs from the Greater Angkor Region and eight other regions in Cambodia were included in this analysis. The eight other regions were specifically mapped for this project. By including areas outside of the Greater Angkor Region, we were able to include varying landscapes and vegetation types in the training dataset. Because the AI model only has access to one snapshot of Bing satellite imagery, we had to clean the data of reservoirs that had previously been mapped within the Greater Angkor Region but were no longer visible in this imagery due to various anthropogenic and natural causes:

- Occlusions: As with many satellite imagery projects, cloud coverage means several known reservoirs are invisible on Bing. Additionally, many reservoirs are largely or completely concealed underneath recent tree cover, and initial experiments confirmed our hypothesis that these were not usable for AI training.

- Temporality and seasonality: As the reservoir mapping in the Greater Angkor Region took place over decades, it became clear that the appearance of some of the reservoirs had substantially changed since the initial mapping. Some, for example, had shrunk, possibly due to changing land-use (see Fig 4). Seasonal differences could explain other changes in the remaining water level: the Bing imagery was probably taken in a different season than the data used for the initial mapping. When we have traditionally conducted this mapping work at Angkor, we have used all available imagery, and, notably, sometimes a reservoir is visible in earlier imagery but not in later imagery or only visible during the wet season, et cetera. As such, many of the reservoirs from the sections within the Greater Angkor Region had to be removed from our training dataset.

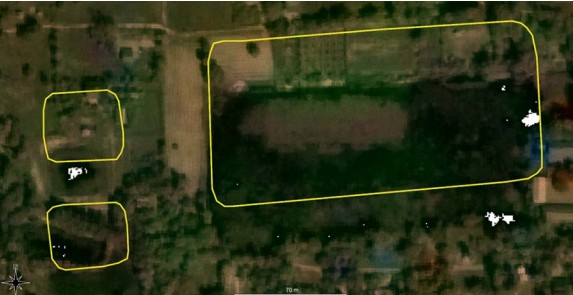

**Fig 4. Three reservoirs whose mapped outline (in yellow) does not correspond to the imagery due to changes in climate and land use conditions.** Image sources: Basemap: Planet Labs, copyright: see section 7 below, overlay: authors.

- Development: Cambodia is a society with a fast-evolving economy. As a result, several reservoirs were found to have been destroyed or built over by modern structures (Fig 5).

All reservoirs were used to create segmentation masks as input for the training process: imagery of each reservoir, including a margin of 50 m to capture its surroundings, was downloaded from Bing. We then created masks by rasterizing each reservoir shape. Due to processing memory limitations, each reservoir image and mask was cut into 512x512 pixel patches.

For the reasons given above and based on inspecting samples of the data, we concluded that several thousand of the generated masks were likely to be unsuitable for input for our ML training. A manual re-mapping of their boundaries to correspond to their current state, as shown in recent satellite imagery, clearly exceeded the project team's resources. To overcome this obstacle, we employed an artificial intelligence technique from the semi-supervised learning (SSL) domain, namely pseudo-labeling [44], to clean our dataset automatically. More specifically, we manually re-inspected a small portion of the reservoirs (around 1 000) from the portion of the data from the Greater Angkor Region and redrew their shapes if needed. This subset of the data was then used to train an initial neural network (see next section for details). Subsequently, all other reservoirs were presented to this neural network, and their reservoir mask prediction was obtained. If a predicted mask and the reservoir shape from the mapping had a reasonably good overlap (measured by their intersection-over-union or IoU metric being above 85%), we concluded that the features of this reservoir were sufficiently similar to the manually curated data, and it was included in the AI training described below. If not, it was sorted out. This quality control process yielded some 3 600 reservoirs for training from the Greater Angkor Region, a much larger number than the 1 000 that were inspected manually, thus allowing for much better regularization and more accurate prediction results.

### 3.2. Training the deep learning model

The resulting data was then used to train a neural network (NN). We used a modern state-of-the art DeepLabV3+ architecture [45] with a ResNet-101 backbone based on the Fast.ai library [46] with PyTorch. The training dataset was split into training and validation subsets with an 80/20 ratio. Various data augmentation techniques were applied, ranging from randomly flipping training images to slightly changing their brightness. Of particular benefit was the random erasing data technique [47], where a few randomly chosen patches of an image are replaced with random noise, thus contributing to better regularization. Empirically, we also found that combining dice and focal loss (as applied in [48]) as the loss function for the NN training boosted prediction results. This setup enabled the model to train for 42 epochs (or iterations) without overfitting.

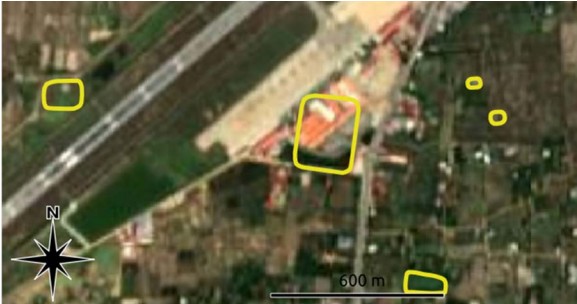

**Fig 5. A mapped reservoir that has disappeared due to modern construction.** Image sources: Basemap: Sentinel-2 cloudless, copyright: see section 7 below, overlay: authors.

The key benchmark tracked during training was IoU, a metric often used for semantic segmentation tasks. With the validation dataset, the model reached an IoU of 0.917. This relatively good value should not be misinterpreted, however, as results for imagery outside the training data regions were much lower (see below).

This training process resulted in a semantic segmentation model that can input any satellite imagery patches of 512 x 512 pixels and compute a corresponding segmentation mask. This resulting data shows the confidence score (between 0% and 100%) for each pixel in this patch that is part of an Angkorian reservoir. Since we were mainly interested in the results for larger areas, we split an area into a grid of patches with appropriate dimensions and then merged the NN prediction results. Finally, we disregard pixels with a confidence score below 50% and used conventional image processing algorithms to merge adjacent remaining pixels into a polygon shape for more straightforward analysis. Fig 6 shows an example result: Reservoir shapes as predicted by the NN are shown in blue. Human-mapped shapes are given in yellow. The NN missed two small reservoirs in the southwest of the image.

## 4. Results

To assess the prediction quality of the artificial intelligence model, we used a two-stage approach: First, we selected eight test areas (with a total size of 43 km²) from regions across the country (Fig 7). Second, to evaluate the impact of substantial differences in the underlying landscape and to assess the prediction quality on a regional or large-scale level, we selected an area of 130 km² within the Greater Angkor Region to test against human reservoir identifiers using only the imagery available through Bing (Fig 8). The latter qualification is important. The workflow for manually mapping reservoirs normally involves analyzing all available imagery from a region, providing time depth and seasonal coverage instead of relying on a single set of imagery. In this experiment, however, human workers were subject to the same restrictions as the AI algorithm. As a result, we can expect the human experts to have performed less effectively than they would under regular conditions.

### 4.1. Test one: Model performance

Our initial experiments indicated that the AI model could not accurately delineate reservoir boundaries (Fig 6 and 9). However, many AI detections did overlap with known reservoirs. Therefore, for evaluation purposes, we defined a reservoir as correctly identified by the model (a True Positive or TP) if the predicted shape overlaps with the mapped reservoir. All other

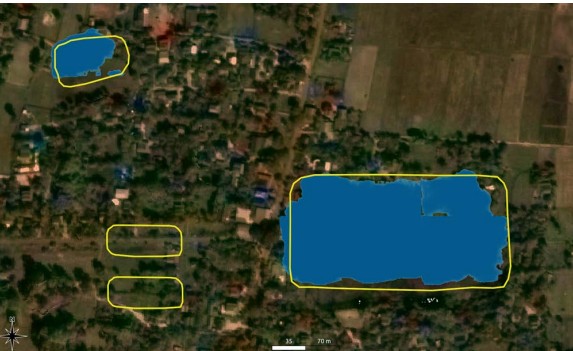

**Fig 6. Example output of the neural network (in blue) alongside human-mapped reservoirs (in yellow).** Image sources: Basemap: Planet Labs, copyright: see section 7 below, overlay: authors.

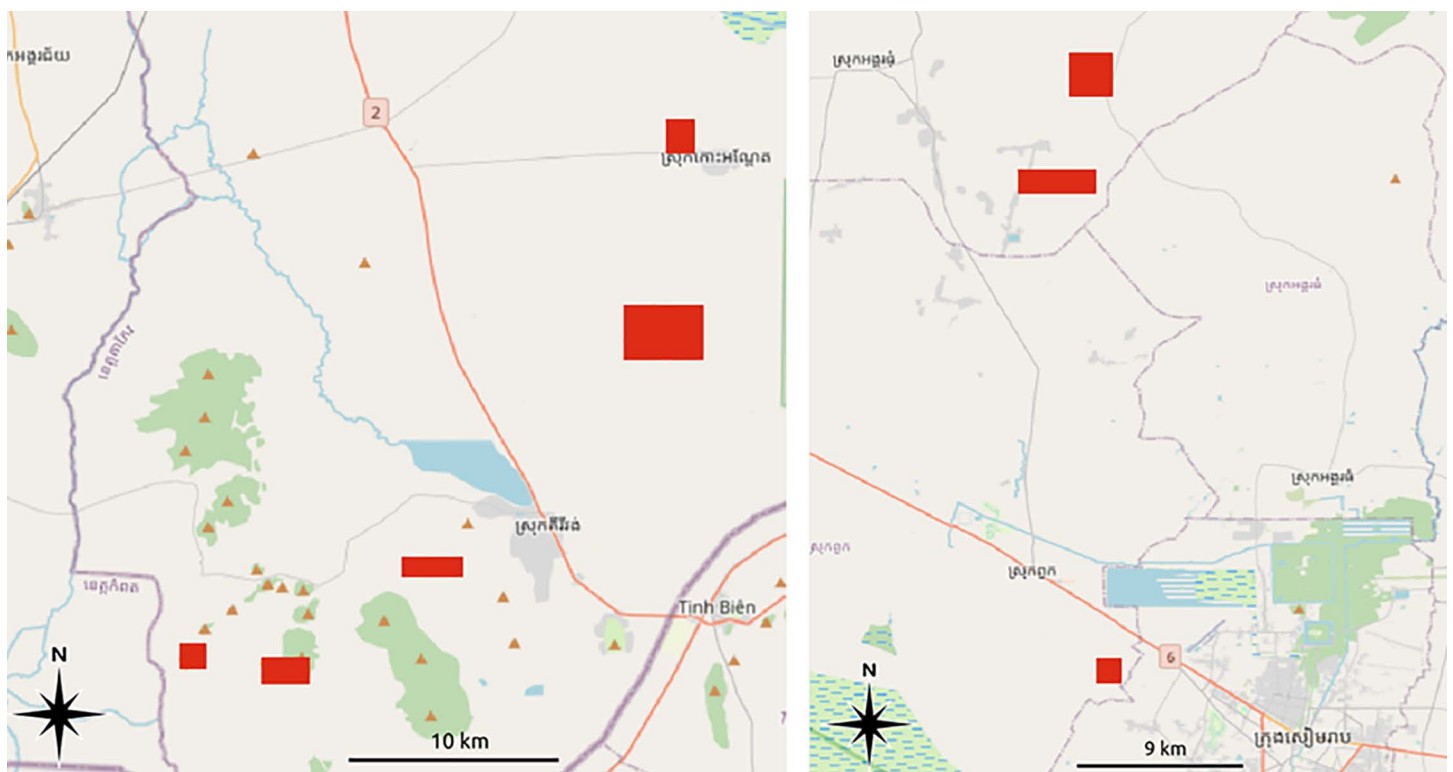

**Fig 7. Test areas (red rectangles, total 43 km²) in Southern Cambodia (left) and near the Greater Angkor region (right).** Basemap: OpenStreetMap.

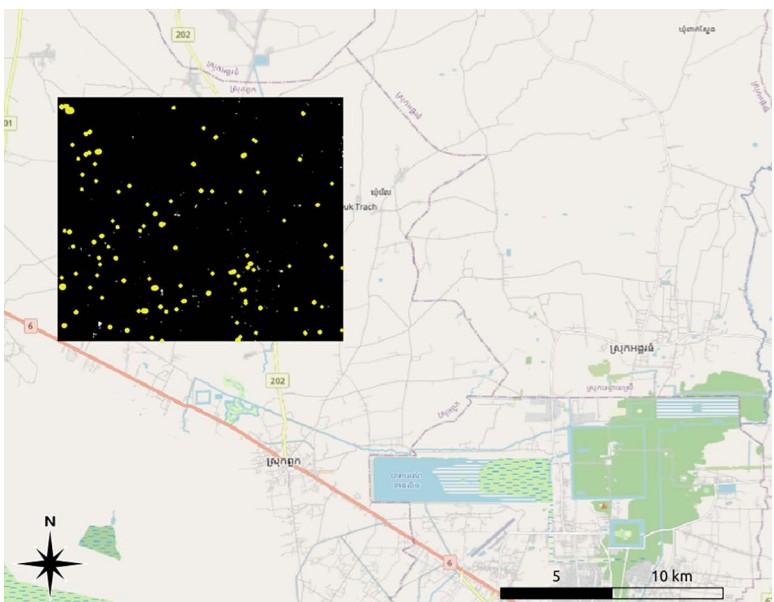

**Fig 8. The test area (in block outline) within the Greater Angkor Region, showing known reservoirs in yellow.**
Image sources: Basemap: OpenStreetMap, overlay: authors.

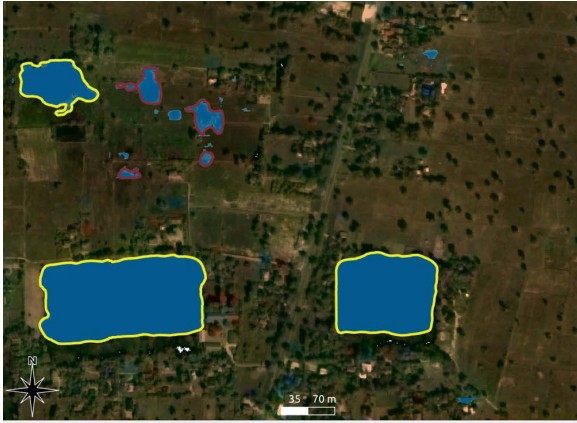

**Fig 9. Example of predictions in GAR test region.** The blue shapes represent the model's predictions overlapping with known reservoirs (yellow borders), while the red borders indicate predictions not within known reservoirs and counted as false positives. Blue areas without borders are disregarded due to their small size. Image sources: Basemap: Planet Labs, copyright: see section 7 below, overlay: authors.

predicted shapes were counted as false positives (FP). While false negatives (FN) are known reservoirs that the model failed to detect, true negatives (TN) cannot be quantified, as they comprise the entire remaining landscape within the test area. In Table 1 below, apart from reporting Precision, and Recall as quality metrics, we also report the $F_1$ score, meaning the harmonic mean of the Precision and Recall, as it takes the imbalanced nature of the data into consideration: reservoirs cover only a small fraction of the Cambodian landscape. The $F_1$-score also illustrates the human effort still needed for expert verification of AI results: even if an AI model detects almost all the reservoirs within a given area, its practical utility would be limited unless the number of FPs is sufficiently low to be manually verified within a reasonable timeframe. As such, the higher the $F_1$-score, the greater the model's utility for archaeologists.

The overall $F_1$ score is around 31% but shows a rather high variance interval, with $F_1$ ranging between 15% and 45% (Table 1). This can partly be explained by the different sizes of the test regions (containing 9 to 71 mapped reservoirs) but also points towards differences in the detection quality per region. This aligns with an issue well documented in the literature: the so-called model drift problem [49]: Applying neural networks to data (satellite imagery in this case) that is even slightly different from the original training data often results in a greatly reduced detection quality. This is most likely caused by variations in the landscape

**Table 1. AI prediction results of the eight test areas, sorted by $F_1$ score (ascending).**

| Area ID | Mapped reservoirs | Predicted by AI | TP | FP | FN | TN | Precision = TP/(TP+FP) | Recall = FP/(TP+FN) | $F_1$ score |
|---|---|---|---|---|---|---|---|---|---|
| 1 | 9 | 17 | 2 | 15 | 7 | Not applicable | 0.12 | 0.22 | 15% |
| 2 | 38 | 12 | 4 | 8 | 34 | | 0.33 | 0.11 | 16% |
| 3 | 30 | 5 | 4 | 1 | 26 | | 0.80 | 0.13 | 23% |
| 4 | 71 | 58 | 17 | 41 | 54 | | 0.29 | 0.24 | 26% |
| 5 | 12 | 8 | 3 | 5 | 9 | | 0.38 | 0.25 | 30% |
| 6 | 21 | 23 | 7 | 16 | 14 | | 0.30 | 0.33 | 32% |
| 7 | 22 | 23 | 8 | 15 | 14 | | 0.35 | 0.36 | 36% |
| 8 | 71 | 57 | 29 | 28 | 42 | | 0.51 | 0.41 | 45% |
| Total | **274** | **203** | **74** | **129** | **200** | | **0.36** | **0.27** | **31%** |

or, more precisely, by a different stochastic distribution of the color values found in the data. Various approaches to deal with this problem have been published [50] but exceed the scope of work for the study presented here. Other obvious FPs sources were modern water-bearing structures (Fig 10) or occlusions caused by clouds.

The model correctly identified 74 of the 274 known reservoirs across the eight test areas, a success rate (recall) of 27%. Of the 203 predicted reservoirs, 129 (64%) were FPs. Considering the difficulties with the dataset outlined above, we consider this result acceptable. While it could not, in its current form, act as a complete substitute for human experts, the result indicates that AI could provide a substantial degree of assistance to manual mapping work.

## 4.2.  Test two: The model vs. the (wo)man

To assess the AI model's viability compared to trained archaeologists (two experts who have, combined, been working in the area for over two decades and two students who were trained by the experts over six months), we selected a section of the Greater Angkor Region (GAR) of approximately 130 km² (see Fig 8). The experts first worked alone to identify visible reservoirs in the Bing satellite imagery, then worked together to create a "final" list of visible reservoirs. Identifying archaeological features in satellite imagery is an iterative process, which was made more difficult for the human experts in this instance by the limitation of using only a single set of satellite imagery. As a result of this limitation, neither expert alone could identify every reservoir, and each had several errors in their initial sweep. The difficulties of manual identification of reservoirs will be addressed further in the discussion section.

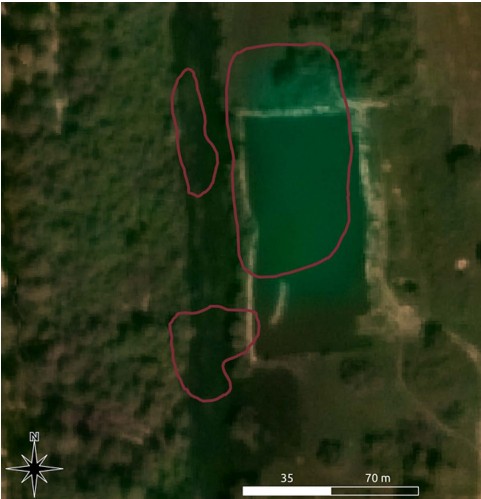

**Fig 10.  False positives caused by modern water-bearing structures, with predictions shown in red.** Image sources: Basemap: Planet Labs, copyright: see section 7 below, overlay: authors.

**Table 2.  Prediction results for human experts and the AI model for the GAR test area.**

|  | Reservoirs | Predicted | TP | FP | FN | TN | Precision | Recall | F₁ score | Time (hrs) |
|---|---|---|---|---|---|---|---|---|---|---|
| Expert 1 | 484 | 539 | 452 | 87 | 32 | Not applicable | 0.84 | 0.93 | 88% | 0:59 |
| Expert 2 | 484 | 415 | 250 | 165 | 234 |  | 0.60 | 0.52 | 56% | 0:11 |
| Student 1 | 484 | 507 | 190 | 317 | 294 |  | 0.37 | 0.39 | 38% | 0:55 |
| Student 2 | 484 | 1427 | 362 | 1065 | 122 |  | 0.25 | 0.75 | 38% | 3:36 |
| **AI model** | **484** | **253** | **95** | **158** | **389** |  | **0.38** | **0.20** | **26%** | **0:02** |

The students were then asked to perform the same task. Their results, as well as those of the individual experts and that of the AI model, were then compared with the "final" results of the two experts combined (Table 2). Expert 1 demonstrates a high recall of 93% and an $F_1$ score of 88%. Expert 2, despite a lower recall of 52%, reaches a good $F_1$ score of 56%. Students 1 and 2 display varying performance degrees, but their $F_1$ score is the same at 38%. While achieving a relatively low recall of 19%, the AI model attains an $F_1$ score of 26%. Taking its score of between 15% and 45% in the other test areas into consideration, its performance could be compared to a student with a somewhat lower degree of experience. One should also note that defining the "final" or ground truth list of visible reservoirs inevitably induces bias - a phenomenon often observed when archaeologists deal with the ambiguous nature of site classification [51].

This bias means that the concrete $F_1$-scores should be treated with caution and most likely contain a high degree of variance. Incorporating the time taken by each entity alongside their performance metrics, it is clear that Expert 1, despite spending 59 minutes, achieved an impressive $F_1$-score, indicating thoroughness and effectiveness in reservoir identification. Expert 2, with only 11 minutes spent, performed fairly efficiently despite the limited time invested. Student 1 spent 55 minutes, suggesting a reasonable proficiency within the given timeframe. In comparison, student 2 invested a substantial 3 hours and 36 minutes and had a high number of FPs, indicating potential inefficiencies or challenges in completing the task. Unsurprisingly, the AI model completed the task in the shortest span of time, requiring only 2 minutes.

## 5. Discussion

As with all automated detection projects mentioned in the introduction, the results delivered by the AI model described above cannot (and should not) yet be used without human expert verification [26]. Typically, its results are comparable to a student's, identifying approximately one-quarter to a third of the reservoirs present in an area. However, it does so at an acceptable level of false positives, much lower than those of the students in our test. These results suggest that a substantial speed improvement could be achieved in the overall mapping process if AI and human expert workflows are combined: an expert merely verifying the AI results and perhaps only cursorily looking at the rest of the landscape within a test area could substantially bring down time invested per $km^2$, possibly to a small fraction of the original time required [49]. This is especially true because the nature of the archaeological landscape leads the reservoirs to cluster around temples. If the AI correctly identifies one reservoir, the human expert can focus on identifying the other reservoirs of the community and the temple with which they are associated. As such, the time invested by the human expert could be reduced by as much as 90%. This is particularly true when investigating much larger geographic contexts. Experience also shows that human attention quickly deteriorates when asked to scan very large amounts of images over a long period of days or weeks, inevitably resulting in a drop in detection quality. Note that this observation, namely that intertwining the human and AI workflow increases the overall productivity and reduces the effect of human fatigue, has been observed by a variety of earlier publications in the field of archaeological remote sensing [38,52].

Cambodia has a surface area of 181 000 $km^2$, meaning even the fastest student (55 minutes) in the experiment above would take around 1 400 hours or 35 weeks at 40 hrs/week for the suggested pre-analysis to cover the entire country. By contrast, AI could process the same area within a few days of computation without deteriorating the quality of the results. Using the AI's output as input, we presume that experts could increase their productivity by a factor of ten or more. Moreover, as previously mentioned, reservoirs typically appear in clusters. By

taking advantage of this, an additional productivity gain for human experts is within reach by concentrating their work only on AI results and their near vicinity. There is a trade-off between productivity and thoroughness in the strategy proposed here. Still, we estimate that with this workflow, a human expert could analyze all of Cambodia within a more reasonable timeframe than would otherwise be attainable.

We achieve $F_1$ scores between 15% and 45% (averaging 31%) in regions where the AI was trained and 26% in the test region, which is attributed to "model drift" as discussed in the paper. Rather than setting a predefined goal, our approach was to assess how far we could progress given the adversarial nature of the data, including limitations of satellite imagery (e.g., cloud occlusions), substantial noise in the dataset, and changes in the visual appearance of reservoirs due to climate change and urban development. When comparing detection quality, we find our AI model to be on par with a slightly below-average student, but it is much cheaper and can process large areas quickly without fatigue or expense. We conclude that AI is essential for extending human productivity, especially for scanning vast areas at a provincial level, where manual efforts would be prohibitively time-consuming and expensive. The AI workflow should complement the human workflow, with humans focusing only on verifying results rather than reviewing entire regions. This approach is already feasible at current $F_1$ scores, with future improvements dependent on securing funding and obtaining more data.

## 6. Conclusion

The primary goal of this study was to develop an AI model capable of providing a valuable "second opinion" to Khmer landscape archaeologists, thereby accelerating the mapping process and significantly enhancing productivity. This objective has been successfully met, providing a reliable tool for identifying archaeological features in Cambodia.

Future work will involve improving the overall prediction quality and then further automating the mapping of previously unexamined parts of the country. It would also be useful to conduct a study comparing this AI algorithm, student and expert satellite investigations with archaeological sites detected using traditional survey methods. A study from southern Peru suggests that satellite surveys can detect 20% of sites previously found through pedestrian surveys [53]. We posit that the efficacy of satellite surveys will differ based on the specific local archaeological record and our experience has shown that satellite imagery is more effective than pedestrian surveys for detecting Angkorian-period reservoirs, but this remains anecdotal and requires further testing. Another objective is to extend this capability to other characteristic archaeological features in the Angkorian world, such as temples. In fact, original research objective was to find Angkor-period temples in satellite imagery. Unfortunately, this proved too difficult at present because of the variation in size and shape of temple mounds. Instead, we pivoted our search to Angkor-period reservoirs, which are always rectangular in shape, with raised embankments, and thus easier for the algorithm to identify. The identification tool has proven extremely valuable, and we hope to one day be able to define precise reservoir boundaries, which we failed to do in this study. Finally, domain adaptation techniques should be explored to enable the AI to better adapt to landscapes it has not encountered during training.

Integrating AI as a tool for reservoir identification into large-scale mapping projects offers promising opportunities for expediting research. However, it remains crucial to underscore the importance of human validation in verifying results. Enhancing the adaptability of neural networks to diverse landscapes through domain adaptation techniques represents a promising avenue for future research. It will enable AI models to contribute more effectively to archaeological mapping endeavors.

## Acknowledgments

The authors thank the Cambodian government, including the Ministry of Culture and Fine Arts and the APSARA Authority. Administrative support for this project was provided by Leiden University, École française d'Extrême-Orient, and the University of Toronto.

## Author contributions

**Conceptualization:** Jürgen Landauer, Sarah Klassen, Wouter Baernd Verschoof-van der Vaart.

**Data curation:** Jürgen Landauer, Sarah Klassen, Adam P. Wijker, Josine van der Kroon, Alexis Jaszkowski.

**Formal analysis:** Jürgen Landauer, Sarah Klassen, Adam P. Wijker, Josine van der Kroon, Alexis Jaszkowski.

**Funding acquisition:** Sarah Klassen.

**Investigation:** Jürgen Landauer, Sarah Klassen, Alexis Jaszkowski.

**Methodology:** Jürgen Landauer, Sarah Klassen, Wouter Baernd Verschoof-van der Vaart.

**Project administration:** Jürgen Landauer, Sarah Klassen.

**Resources:** Jürgen Landauer, Sarah Klassen.

**Software:** Jürgen Landauer.

**Supervision:** Jürgen Landauer, Sarah Klassen.

**Validation:** Jürgen Landauer, Sarah Klassen, Adam P. Wijker.

**Visualization:** Jürgen Landauer, Sarah Klassen, Adam P. Wijker.

**Writing – original draft:** Jürgen Landauer, Sarah Klassen.

**Writing – review & editing:** Jürgen Landauer, Sarah Klassen, Adam P. Wijker.

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
