## [Decision Letter · Decision Letter 0]

19 Aug 2024

PONE-D-24-26472Beyond the Greater Angkor Region: Automatic large-scale mapping of Khmer Empire reservoirs in satellite imagery using Deep LearningPLOS ONE

Dear Dr. Klassen,

Thank you for submitting your manuscript to PLOS ONE. After careful consideration, we feel that it has merit but does not fully meet PLOS ONE’s publication criteria as it currently stands. Therefore, we invite you to submit a revised version of the manuscript that addresses the points raised during the review process.

The two peer reviewers thought that this paper had merit, as do I, but improvements are recommended. The overall content and presentation requires some evaluation to streamline and present the overall aims, objectives and findings. With respect to the content itself, improvements in method descriptions, data availability and reference citations is in need of consideration. 

We look forward to receiving your revised manuscript.

Kind regards,

Michael D. Petraglia, Ph.D.

Academic Editor

PLOS ONE

 [European Union’s Horizon 2020 Marie Sklodowska-Curie grant agreement No. 896092  (sk)].  

[The authors thank the Cambodian government, including the Ministry of Culture and Fine Arts and the APSARA Authority. This project has received funding from the European Union’s Horizon 2020 Marie Sklodowska-Curie grant agreement No. 896092 and the University of Toronto. Administrative support for this project was provided by Leiden University, École française d’Extrême-Orient, and the University of Toronto. ]

 [European Union’s Horizon 2020 Marie Sklodowska-Curie grant agreement No. 896092  (sk)].  

5. We note that you have indicated that there are restrictions to data sharing for this study. PLOS only allows data to be available upon request if there are legal or ethical restrictions on sharing data publicly. For more information on unacceptable data access restrictions, please see http://journals.plos.org/plosone/s/data-availability#loc-unacceptable-data-access-restrictions. 

6.  We note that 4, 7, 8 ,9 10 and 11 in your submission contain [map/satellite] images which may be copyrighted. All PLOS content is published under the Creative Commons Attribution License (CC BY 4.0), which means that the manuscript, images, and Supporting Information files will be freely available online, and any third party is permitted to access, download, copy, distribute, and use these materials in any way, even commercially, with proper attribution. For these reasons, we cannot publish previously copyrighted maps or satellite images created using proprietary data, such as Google software (Google Maps, Street View, and Earth). For more information, see our copyright guidelines: http://journals.plos.org/plosone/s/licenses-and-copyright.

a You may seek permission from the original copyright holder of 4, 7, 8 ,9 10 and 11 to publish the content specifically under the CC BY 4.0 license.  

b If you are unable to obtain permission from the original copyright holder to publish these figures under the CC BY 4.0 license or if the copyright holder’s requirements are incompatible with the CC BY 4.0 license, please either i) remove the figure or ii) supply a replacement figure that complies with the CC BY 4.0 license. Please check copyright information on all replacement figures and update the figure caption with source information. If applicable, please specify in the figure caption text when a figure is similar but not identical to the original image and is therefore for illustrative purposes only.

Natural Earth (public domain): http://www.naturalearthdata.com

Reviewers' comments:

Reviewer's Responses to Questions

**Comments to the Author**

1. Is the manuscript technically sound, and do the data support the conclusions?

Reviewer #1: Yes

Reviewer #2: Partly

2. Has the statistical analysis been performed appropriately and rigorously?

Reviewer #1: Yes

Reviewer #2: N/A

3. Have the authors made all data underlying the findings in their manuscript fully available?

Reviewer #1: Yes

Reviewer #2: No

4. Is the manuscript presented in an intelligible fashion and written in standard English?

Reviewer #1: Yes

Reviewer #2: Yes

5. Review Comments to the Author

Reviewer #1: The use of artificial intelligence and remote imagery in archaeological research is a burgeoning field, and Landauer et al. provide an exciting example of the ways these relatively new methodological tools can be fruitfully applied. The authors construct a deep learning model using Deeplab V3+ to automate the identification of Angkor period water reservoirs, and report partial success. While the deep learning model is obviously not perfect, the authors contend that it is accurate and precise enough to aid archaeologists in later terrestrial survey and identification of archaeological sites. I ultimately agree with this assessment, and believe that this article would be a welcome addition to the literature with the minor changes below. However, I have a number of concerns which I would like to see the authors address prior to publication.

1. Foremost among these, the GitHub link provided directs only to a singular empty readme file, rather than the full body of the code. While I appreciate that the exact location of the archaeological reservoirs cannot be reported, the full code used in this analysis should be made easily available for replicability – especially considering the methodological nature of this paper.

2. Although the introduction is largely solid, lines 100 to 105 seem out of place. Rather than framing the research problem in question, they detail a failed attempt at a different project. Instead of including this section in the introduction, I might recommend shifting these lines to the discussion or conclusion as an area for future research. The background is similarly well written, with a note that the claim on lines 165-166, “This pattern of more and distinctly less densely occupied agricultural areas seems broadly consistent at the other urban centers surveyed in Cambodia in 2015,” needs a proper citation to the aforementioned survey.

3. The methods section includes material that may be better suited for either the discussion or the introduction. In particular, lines 184 to 205 do not outline the specific methods of this analysis but instead set up important background context. These lines should be moved to the introduction or background. Furthermore, the background section of this paper would benefit from an elaboration of the broader context this work takes place in. How, specifically, do the methods outlined in this paper differ from the works cited between lines 181 to 205? (Zimmer-Dauphinee, VanValkenburgh, and Wernke 2024) conducted similar research in another region; how does their approach differ?

4. Line 330 – The authors mention ‘Precision’ and ‘Recall’ as their quality metrics, but do not describe the exact equation for calculating these metrics.

5. While not necessarily for the acceptance of this paper, I would encourage the authors to consider Snyder and Haas 2024 which examines the efficacy of manual satellite survey. A review and comparison to the model vs. expert portion this study, as presented from lines 368 to 396.

6. At the moment, the discussion and conclusion narrowly situate the work in its relevance to future survey projects towards the Angkor region of Cambodia. While this is important in and of itself, the paper would be significantly improved through broader articulation with the satellite survey and artificial intelligence in the archaeological literature. For example, the authors could discuss similarities and differences between this work and many of the articles cited between lines 187 and 195.

7. Line 225 through 259 detail challenges with Bing satellite imagery. Why was other imagery not used instead? Google Earth allows users to examine imagery from a variety of dates, and has successfully been used in the past to monitor archaeological sites (Contreras and Brodie 2010; Contreras 2010).

Overall, this is a strong article that I look forward to seeing published. However, prior to that, I believe the authors need to situate their methodological approach, its relevance, and implications for future research in a broader context.

References Cited:

Contreras, Daniel A.

2010 Huaqueros and Remote Sensing Imagery: Assessing Looting Damage in the Virú Valley, Peru. Antiquity 84(324). Cambridge University Press: 544–555.

Contreras, Daniel A., and Neil Brodie

2010 The Utility of Publicly-Available Satellite Imagery for Investigating Looting of Archaeological Sites in Jordan. Journal of Field Archaeology 35(1). Taylor & Francis: 101–114.

Snyder, Thomas J., and Randall Haas

2024 Unstructured Satellite Survey Detects up to 20% of Archaeological Sites in Coastal Valleys of Southern Peru. PLOS ONE 19(2). Public Library of Science: e0292272.

Zimmer-Dauphinee, James, Parker VanValkenburgh, and Steven A. Wernke

2024 Eyes of the Machine: AI-Assisted Satellite Archaeological Survey in the Andes. Antiquity 98(397): 245–259.

Reviewer #2: This manuscript reports on the use of a deep learning model to identify ancient reservoir features in the Greater Angkor Region. The article is interesting and well written. I do have a few comments and concerns with the current draft that need to be addressed before proceeding further.

General Comments:

The main issue I find with the current manuscript is that the definition of an "acceptable" result or error rate is not well justified. In some fields, F1 scores less than 0.9 might be considered unacceptable, while others are perfectly fine with F1 scores of 0.5. The results presented in this manuscript are, by most standards, quite low. While this does not mean that the results you present are unusable, the burden is on you to: 1) define what your working definition of an "acceptable" performance metric is and 2) justify the utility by citing its comparison with other modes of identification and/or speed of evaluation. You do get to this latter part later in the manuscript, but don't really define what you deem as "acceptable".

As currently written, someone skeptical of AI who picks up this article will likely see this as another example of why these approaches "aren't worth it" or "don't work well". I think your strongest argument is to further emphasize the fact that with X amount of time to train a model (needs to be quantified), your method performs roughly as well as a student that needed 6+ months of training. As such, the tool is useful for initial exploratory analysis that experts can fine tune and verify much more quickly (similarly to having students assist in data analysis tasks). But with this, a more nuanced discussion of how you are determining acceptability of results is needed.

Other Specific Comments

Data availability: A link to a Github page was provided with the manuscript but there is no code currently uploaded in that location. This should be updated before acceptance. Additionally, there is no mention of this Github link in the manuscript, itself, and should be listed either in the acknowledgements or somewhere in the methods section so readers can find it.

Figures:

Figure 1 and Figure 2 need legends. It isn't clear what the different colors represent.

For Figure 3, either add legend for what the orange boundaries represent or include a description in the caption.

Line by Line Comments:

Line 112: I think a figure depicting the broader region and its different historic boundaries would be helpful here.

Lines 228-231: Do you know when these Bing images were taken? Are they all from the same time period or do they span multiple years or seasons?

Lines 243-244: "Bing images was taken in a different season..." Specify. Which season?

Lines 277-278: Please provide some additional details and justifications for why this architecture and backbone model were selected. Details on how they operate are needed for those unfamiliar with these kinds of methods.

Line 307: Can you indicate this on a map (either in an existing figure or a new one) to show the geographic coverage of the test areas?

Lines 346-347: "...different stochastic distribution of the color values found in the data." This is a particularly significant problem that this work (or future work) should address. Atmospheric corrections to normalize color values might significantly improve results in this case.

Lines 405-407: "These results suggest that a substantial speed improvement could be achieved in the overall mapping process if AI and human expert workflows are combined" This same argument has been made by several other studies as well. For example:

Davis, D. S. (2020). Defining what we study: The contribution of machine automation in archaeological research. Digital Applications in Archaeology and Cultural Heritage, 18, e00152. https://doi.org/10.1016/j.daach.2020.e00152

Zimmer-Dauphinee, J., VanValkenburgh, P., & Wernke, S. A. (2024). Eyes of the machine: AI-assisted satellite archaeological survey in the Andes. Antiquity, 98(397), 245-259.

Line 430: "...providing a reliable tool..." See my general comments, above, but is it an overstatement to call this tool "reliable", given that overall it only achieved an F1 of ~30-40%? This is why a better discussion of what is viewed as acceptable or reliable is needed.

6. PLOS authors have the option to publish the peer review history of their article (what does this mean? ). If published, this will include your full peer review and any attached files.

**Do you want your identity to be public for this peer review?** For information about this choice, including consent withdrawal, please see our Privacy Policy .

Reviewer #1: No

Reviewer #2: No

---

## [Author Response · Author response to Decision Letter 1]

9 Feb 2025

October 21, 2024

Dear Dr. Petraglia,

My co-authors and I would like to thank you and the reviewers again for the comments

and questions on our manuscript. We are happier with the current version and thank the reviewers for their thoughtful comments. Each of the comments, concerns, and questions have been addressed, and the revised materials have been uploaded to the online portal.

Reviewer 1

1. GitHub: We have now made the complete source code available on GitHub. The location is also mentioned in section 7 of the paper.

2. Reviewer 1 made a very good point that lines 100 to 105 are out of place in the introduction. We have moved that information to the conclusion as a suggestion for further research.

3. We have rightly moved the section on Deep Learning Models to the Background section and added the references suggested by Reviewer 1.

4. Reviewer 1 asked for the exact equation for calculating precision and recall. We added the formulas for precision and recall to Table 1

5. We added a reference to the Snyder and Haas 2024 paper as suggested future work in the conclusion section.

6. Reviewer 1 asked us to include a broader articulation with the satellite survey and artificial intelligence in the archaeological literature. We have added several citations throughout the paper to improve our discussion of the broader potential uses for this method.

7. Reviewer 1 asked why we used Bing imagery rather than other satellite imagery, including Google Earth. The critical advantage of Bing was that it is freely available (if certain usage restrictions were met). By contrast, Google Earth, which, to our knowledge, has comparable data quality, also has a substantial fee outside the means of this project. We have updated the text: " To train our ML model, we used satellite data of Cambodia from Microsoft Bing, which provides high-resolution imagery of up to 30 cm per pixel at low or no cost."

Reviewer 2

1. F1-scores: We agree with Reviewer 2 that a more nuanced approach to addressing the F1 score of this study is warranted. Our lead author works in the automotive industry, and in the field of autonomous vehicles, any F1 score below 0.99 is unacceptable, as human lives are possibly at stake. This is not the case in our archaeological study, and our core argument and metric do not evolve around any specific F1 threshold but are the relative cost (or human effort). The F1 score is merely used (and reported) to indicate whether we have reached an acceptable balance between True and False Positives. We have expanded on this discussion in the discussion section.

2. We have uploaded the source code to GitHub and referenced it in text.

3. We have added legends to figures 1, 2, and 3.

4. Reviewer 2 asked for a figure depicting the broader region and its different historical boundaries. Unfortunately, this is exceedingly difficult to do as the data is unavailable and constantly changing, and creating such a map would be beyond this project's scope. We hope to be able to produce such maps after further research and mapping in the area. Much of this is underway as part of a series of Lidar acquisitions conducted in 2023 and 2024.

5. Unfortunately, Bing does not disclose when the images were taken or if they span multiple years or seasons. We found they are the most recent and highest-quality datasets that Bing can amass. Bing was chosen because it is essential cost-free. By contrast, Google Earth and other data providers offer time stamps, but the cost of their services greatly exceeds the budget of this project. When working on a new region map, we will look at all available imagery of a given area. Ideally, we would have access to as much imagery as possible to run this model and human verification. However, because this paper focused primarily on the method and the human testers were also only given access to the Bing imagery, we determined that it was suitable for our needs. Moving forward, we will focus on using this method (in addition to manual identification and pedestrian surveys) to create maps for otherwise unmapped areas of Cambodia. In that case, running the models on all available imagery will be highly useful. We may also find that the model works better under some conditions (e.g., seasons) than others.

6. Similarly, when we mention that the “Bing images were taken in a different season,” we cannot specify what season because we do not know (for reasons discussed above). However, variations in color distribution suggest differences in vegetation and, thus, seasons. As such, we changed the text into "Bing imagery was probably taken…"

7. Reviewer 2 asked us to provide additional details and justification for why this architecture and backbone model were selected and how they operate. We now mention the origin of DeeplabV3+ (from Google since 2015) in section 3.2 and that it outperforms other architectures in this regard. However, we think explaining its unique features (such as atrous separable convolutions and atrous spatial pyramid pooling) in an archaeology paper would be inappropriate and refer the reader to the corresponding Chen et al. paper.

8. We have now inserted a new figure showing a map of the test areas (fig. 8).

9. The differences in the stochastic distribution of the data are one of the key issues studied in the model drift (or domain shift) literature. This is a "hot" research topic and is being studied in both aerial (satellite, aircraft-based) studies and ground-based (e.g., autonomous driving) papers. To our knowledge, atmospheric corrections have already been studied intensively in the industrial research domain. We modified our text to point readers toward our citation 48, which gives a good and recent overview of the state of research in that field.

10. We have added references to Zimmer-Dauphinee et al. 2024 and Davis 2020, who also note that substantial speed improvements could be achieved in the overall mapping process by combining AI and human workflows.

11. We have added a section of the text to discuss why a relatively low F1 score is acceptable for this type of work.

Academic Editor

1. The academic editor noted that the overall content and presentation require some evaluation to streamline and present the overall aims, objectives, and findings and that we should improve the methods description, data availability, and reference citations. We hope that the above revisions have achieved this.

Journal Requirements

1. We updated the formatting of the draft to meet PLOS ONE’s style requirements.

2. We have moved the funding information from the acknowledgments to the financial disclosures section.

3. We have noted that the site locations used for training and testing may be available for research purposes, with appropriate request to the authors. The data are not publicly available due to ethics of archaeological site protection. The data is available for download with permission at tdar.org (tDAR id: 501902).

4. We have switched the satellite imagery to Planet.com, which we believe adheres to the copyright requirements.

Please let me know if you or the reviewers have additional questions or concerns. Again, we thank you for this opportunity.

Sincerely,

Dr. Sarah Klassen

---

## [Editor Report · Decision Letter 1]

19 Feb 2025

Beyond the Greater Angkor Region: Automatic large-scale mapping of Khmer Empire reservoirs in satellite imagery using Deep Learning

PONE-D-24-26472R1

Dear Dr. Klassen,

We’re pleased to inform you that your manuscript has been judged scientifically suitable for publication and will be formally accepted for publication once it meets all outstanding technical requirements.

Kind regards,

Michael D. Petraglia, Ph.D.

Academic Editor

PLOS ONE
---

## [Editor Report · Acceptance letter]

PONE-D-24-26472R1

PLOS ONE

Dear Dr. Klassen,

I'm pleased to inform you that your manuscript has been deemed suitable for publication in PLOS ONE. Congratulations! Your manuscript is now being handed over to our production team.

Kind regards,

on behalf of

Professor Michael D. Petraglia

Academic Editor

PLOS ONE